# Emerging Roles of C-Myc in Cancer Stem Cell-Related Signaling and Resistance to Cancer Chemotherapy: A Potential Therapeutic Target Against Colorectal Cancer

**DOI:** 10.3390/ijms20092340

**Published:** 2019-05-11

**Authors:** Mohamed Elbadawy, Tatsuya Usui, Hideyuki Yamawaki, Kazuaki Sasaki

**Affiliations:** 1Laboratory of Veterinary Pharmacology, Department of Veterinary Medicine, Faculty of Agriculture, Tokyo University of Agriculture and Technology, 3-5-8 Saiwai-cho, Fuchu, Tokyo 183-8509, Japan; mohamed.elbadawy@fvtm.bu.edu.eg (M.E.); skazuaki@cc.tuat.ac.jp (K.S.); 2Department of Pharmacology, Faculty of Veterinary Medicine, Benha University, Moshtohor, Toukh, Elqaliobiya 13736, Egypt; 3Laboratory of Veterinary Pharmacology, School of Veterinary Medicine, Kitasato University, Towada, Aomori 034-8628, Japan; yamawaki@vmas.kitasato-u.ac.jp

**Keywords:** c-Myc, cancer organoid, colorectal cancer, cancer stem cell

## Abstract

Myc is a nuclear transcription factor that mainly regulates cell growth, cell cycle, metabolism, and survival. Myc family proteins contain c-Myc, n-Myc, and l-Myc. Among them, c-Myc can become a promising therapeutic target molecule in cancer. Cancer stem cells (CSCs) are known to be responsible for the therapeutic resistance. In the previous study, we demonstrated that c-Myc mediates drug resistance of colorectal CSCs using a patient-derived primary three-dimensional (3D) organoid culture. In this review, we mainly focus on the roles of c-Myc-related signaling in the regulation of CSCs, chemotherapy resistance, and colorectal cancer organoids. Finally, we introduce the various types of c-Myc inhibitors and propose the possibility of c-Myc as a therapeutic target against colorectal cancer.

## 1. Introduction

Myc is a family of regulator genes and proto-oncogenes that encodes essential nuclear transcription factors and belongs to the superfamily of basic helix-loop-helix (b-HLH) DNA-binding proteins. Myc mainly regulates cell growth, proliferation, differentiation, cell cycle, metabolism, survival, apoptosis, as well as tumorigenesis [1,2,3,4,5,6,7]. In mammals, Myc family proteins contain c-Myc, n-Myc, and l-Myc. They possess an identical general topography because of the high similarity of their genome, ribonucleic acid, and protein sequences. First, c-Myc was recognized as an oncogene that is accountable for avian leukemia and sarcoma [8,9]. Shortly after that, n-Myc and l-Myc were found in neuroblastoma and small lung cell carcinoma [10,11]. Since the identification of the characterization of c-Myc in Burkitt lymphoma [12,13], several articles have been disseminated that discuss the pathophysiological roles of Myc family proteins in various cancer [14,15,16,17]. Deregulation of c-Myc is involved in genome instability and tumorigenesis, and sustained the growth of tumors [18,19]. Thus, the inhibition of c-Myc is a promising therapeutic strategy for human cancer [20]. Nevertheless, there remain several points that are currently poorly understood about the complex roles of c-Myc in tumor cells. In this article, we introduce the novel information on the structure and various physiopathological functions of c-Myc and discuss the therapeutic potentials of c-Myc in colorectal cancer.

## 2. Structures of Myc Family Proteins

The mammalian Myc protein consists of a transcriptional activation domain, a central region, a nuclear localization sequence (NLS), and an area involved in DNA binding (Figure 1A). A human c-Myc gene is situated on chromosome 8, band q24.21 and includes three exons plus two introns [21]. c-Myc is a 65 kDa nuclear phosphoprotein, which belongs to a basic helix-loop-helix leucine zipper (b-HLH-LZ) protein family [20,22]. c-Myc includes 439 amino acids (aa) and constitutes the N-terminal transactivation domain (NTD), the central area, and the C-terminal domain (CTD) (Figure 1B).

NTD is the primary regulatory region of c-Myc and is mainly involved in transactivation and transrepression [23]. Within NTD, there is one transcription activation domain (TAD), and there are three ~20 amino acid segments termed Myc box-I, II, and III (MBI, MBII, and MBIII), which are essential for the various biological functions of c-Myc [24,25].

MBI and MBII are situated at aa 45–63 and 129–143, respectively, and are accountable for the regulation of transcription and transformation. Although MBI is required for gene activation and protein degradation, deletion of this region only partially abolishes the transforming ability of c-Myc [23,26]. MBII is critical for the most functions of c-Myc [23,27] and is necessary for c-Myc to transform, drive cell proliferation, inhibit differentiation, repress gene transcription, and activate several target genes [28]. MBIII mediates apoptosis, transformation, transcriptional repression, and lymphomagenesis [29].

CTD is essential for DNA binding and protein interactions. CTD involves a basic region that binds to CACGTG E-box and a helix-loop-helix leucine zipper domain (HLH-LZ), necessary for c-Myc to bind with its primary correlative protein, Myc associated protein X (MAX) (Figure 1B) [2,30,31].

Recently, it was reported that the central region of c-Myc is assumed to be a critical part in oncogenic transformation through the regulation of apoptotic responses [29]. Nevertheless, the fundamental molecular pathways are still unknown. In the central region, MBIV is involved not only in apoptosis and transformation but also in modulation of DNA-binding [32].

## 3. General Functions of c-Myc

c-Myc is involved in the control of several normal cellular functions including growth, proliferation, differentiation, progression of cell cycle, maintenance of cell size, maintenance of genomic integrity, angiogenesis, and apoptosis (Figure 2) [7,33,34,35,36]. Expression of c-Myc is usually regulated by growth factors and extracellular matrix contacts. In quiescent cells, the expression level of c-Myc is low. Once there is entry of the cell into the cell cycle, the expression of c-Myc rapidly increases. Then, it declines to remain at the basal expression level in cycling cells [28,37,38].

It is also known that c-Myc gene expression is the highest during embryonic stages and is downregulated in mature organs [39]. In normal cells, cell cycle checkpoint proteins regulate c-Myc expression. In c-Myc^−/−^ mice, an embryonic lethal event occurs—which is associated with major defects in angiogenesis, proliferation, and differentiation [35,40]—due to the lack of proper vascular endothelial growth factor signaling [35]. Embryos lacking n-Myc are subject to neuroectodermal and heart teratogenic changes between E10.5 and E11.5 [41], whereas l-Myc knockout (KO) mice show no apparent defects [42]. In transgenic mouse models with restricted expression of c-Myc, cell proliferation was arrested, and the cell cycle checkpoint proteins were stimulated through the regulation of p53, alternate open reading frame (ARF), bcl-2-like protein 11 (BIM), and phosphatase and tensin homolog deleted from chromosome 10 (PTEN) that cause cell growth arrest or death [43,44,45].

## 4. Roles of c-Myc in Cancer

c-Myc regulates many important biological pathways that are involved in the growth and proliferation of tumor cells. Deregulation of c-Myc is involved in several pathological conditions, including genomic instability, uncontrolled cell proliferation, immortalization, escape from immune cells, angiogenesis, and metastasis (Figure 2) [14,15,16,17]. In diffuse large B-cell lymphoma, the frequency of c-Myc transformations is 33.3% at the DNA level and 16.1% at the protein level [46,47]. It has been reported in breast cancer that c-Myc amplification is recognized in about one-half of breast cancer susceptibility gene (BRCA)1-mutated tumors, as compared with approximately 20% in sporadic tumors [48]. The Cancer Genome Atlas (TCGA) database shows that 28% of all cancer patients have the genetic abnormalities in at least one of the Myc family composed of c-Myc, n-Myc, and l-Myc [49].

Moreover, c-Myc organizes different cancer cellular functions, including cell cycle, cellular proliferation, survival, and metabolic reprogramming [5,15,17,50,51,52]. Disruption of the strict regulation of c-Myc expression results in increased expression of several genes and uncontrolled cell proliferation, which increase the risk of acquiring secondary mutations contributing to tumor development. Since deregulation of c-Myc occurs in more than half of all cancers and usually correlates with aggressiveness, resistance to therapy, and poor prognosis, it is necessary to find ways to target this pathway.

## 5. Roles of c-Myc in Colorectal Cancer

Colorectal cancer (CRC) is the most common gastrointestinal malignancy with more than 1,800,000 newly diagnosed cases and more than 860,000 deaths in 2018 [53]. Sequence analysis reveals that each tumor genome has numerous mutations, which deregulate signaling pathways that control the growth and survival of colon epithelial cells [54]. Recurrent genetic alterations of Wnt and Ras signal transduction pathways occur in the majority of colorectal cancers [54]. Despite their genomic heterogeneity, a significant upregulation of c-Myc proteins—a downstream effector of both Wnt-and Ras-dependent signal transduction—is common in 70% of CRC, suggesting the important role of c-Myc in CRC [54,55,56].

c-Myc is a transcription factor that regulates a gene expression, which determines the characteristics of epithelial stem cells of colon tissues [16,57]. Furthermore, genetic ablation of c-Myc suppresses intestinal tumorigenesis in mouse models of CRC, indicating that c-Myc function is essential for colorectal tumorigenesis [54,58].

## 6. Roles of c-Myc in the Regulation of Cancer Stem Cell-Related Signaling

Cancer stem cells (CSCs) are capable of self-renewal and differentiation into heterogeneous tumor cells, and they have been considered to be responsible for tumor initiation, proliferation, and recurrence [59,60]. Additionally, CSCs are resistant to chemotherapy and radiotherapy and are associated with cancer metastasis [61]. Since CSCs can survive after traditional chemotherapy and result in tumor recurrence and drug resistance [62,63], elimination of CSCs in tumor tissues may become an effective anti-cancer therapeutic strategy. To achieve this goal, significant efforts have been performed to explore the signaling mechanisms underlying the self-renewal and the differentiation of CSCs, as well as the development of regimens targeting the CSCs. In particular, Wnt/β-catenin, Hedgehog, and Notch pathways are implicated in playing the major roles in the regulation of the functions of CSCs.

The denomatous polyposis coli gene (APC), a tumor suppressor gene, is usually downregulated in the CRC and is known to be a central hub in early CRC [64]. Mutations in the APC gene often lead to altered β-catenin regulation via the Wnt signaling pathway [65,66,67]. When the Wnt signaling pathway is deactivated, a low cytoplasmic level of β-catenin is maintained. Contrarily, when the Wnt signaling pathway is activated, the concentration of β-catenin increases in the cytoplasm and migrates to the nucleus, where it serves as transcriptional machinery for c-Myc and cyclin D1 (Figure 3) [68,69]. The prognostic value of Wnt signaling for CRC patients remains debatable. Recent studies reported that CRC patients with marked Wnt and c-Myc signaling activation by gene expression-based CRC classifications show better prognosis (superior survival after relapse) [70,71]. Another study also reported that overexpression of c-Myc protein is significantly correlated with better survival of CRC patients [72]. Contrarily, a meta-analysis revealed that the accumulation of nuclear β-catenin could be a biomarker for late phase and worse survival of CRC [73].

In CRC, mutations of Wnt pathways induce transactivation of T cell factor-4 (TCF-4) target genes, a main transforming episode in CRC. Perturbation of β-catenin/TCF-4 activity in CRC cells produces a prompt G1 arrest and stops a physiologically active genetic program in the proliferative compartment of colon crypts [57]. c-Myc gene is targeted in this signaling pathway, as it plays a major role in this switch by direct repression of the p21 promoter. After disruption of β-catenin/TCF-4 activity of CRC cells, the declined expression of c-Myc results in the transcription of p21, which in turn triggers G1 arrest and differentiation. Therefore, the β-catenin/TCF-4 complex represents the major switch that regulates proliferation and differentiation in healthy and malignant intestinal epithelial cells (Figure 3) [57].

Activation of Notch signaling can upregulate many signaling pathways that favor CRC cell survival [74]. Pathways that are activated by *Notch* signaling include PI3K/AKT signaling [75], c-Myc, and epithelial growth factor receptor (EGFR). A recent study revealed that the co-activation of the Notch pathway by mastermind-like protein (MAML)-1 could transcriptionally bind to cyclin D1 and c-Myc promoters in CRC cell lines [76]. As cyclin D1 and c-Myc are closely associated with cell cycle progression, the anti-cancer effect of Notch inhibitors might be related to its inhibitory effect on cell cycle progression [77,78,79].

It has also been demonstrated that the self-renewal ability of colorectal CSCs is dependent on the activity of Hedgehog/Glioma-associated oncogene homolog GLI-1 pathway in vivo via influencing CRC progression by connecting to the Wnt pathway [80]. GLI-1 can decrease the cellular concentration of β-catenin, the expression of c-Myc, and the proliferation ability [81]. These results suggest that targeting c-Myc might achieve significant therapeutic efficacy in CRC.

## 7. Roles of c-Myc in Resistance of Chemotherapy

Nowadays, cancer therapy mainly depends on conventional chemotherapeutic agents, which mostly cause cytotoxicity in rapidly proliferating cells by damaging DNA. However, cancer cells often become refractory to chemotherapeutic agents. The phenomenon known as chemoresistance is due to either intrinsic factors or acquired ones after genotoxic therapy [82]. The mechanisms by which tumor cells increase chemoresistance could be attributable to an increase in the machinery of cellular DNA repair [83,84], suppression of apoptosis [85,86], and enhanced efflux of genotoxic drugs [87,88]. A major impediment to cancer chemotherapy is the emergence of multi-drug cross-resistance. Once tumor cells get resistant to conventional chemotherapy such as cisplatin, it naturally develops cross-resistance to many other chemotherapeutic agents, leading eventually to the failure of treatment in more than 90% of patients [87,89]. Therefore, the main clinical strategy for effective chemotherapy is the fast eradication of tumor cells before the emergence of chemoresistance.

The increase of c-Myc expression may induce activation of some downstream genes, which triggers the cell cycle with DNA synthesis and eventually launches genomic instability [90,91,92]. The genomic instability due to c-Myc overexpression is often accompanied by various chromosomal abnormalities, such as the presence of extrachromosomal elements [93], centromere and telomere fusions [94], chromosome and chromatid breaks, deletions, translocations, aneuploidy, and inversions [92,95,96,97]. The aggregation of genomic instability brings the vulnerability of tumor cells to DNA-damaging agents (Figure 4) [98,99].

It is recognized that a higher level of c-Myc is closely correlated with chemotherapy resistance in many cancers, such as prostate cancer, ovarian cancer, melanoma, lung cancer, and hepatocellular carcinoma [100,101,102,103]. In patients with ovarian cancer, high expression levels of c-Myc are associated with cisplatin resistance, tumor recurrence, and poor overall survival [104].

Moreover, it was reported that expression of c-Myc in surviving tumor cells increases after treatment with platinum-based chemotherapy [101]. c-Myc protects embryonal rhabdomyosarcoma cell lines from radiation-induced apoptosis and DNA damage as well as promotes the radiation-induced DNA repair [105]. Considering these reports, c-Myc seems to be a promising target to overcome chemoresistance in various types of cancers. It was also reported that the silencing of c-Myc by small-interfering (si)RNA boosts cisplatin sensitivity in cisplatin-resistant melanoma cell lines in vitro and inhibits tumorigenesis in xenograft models of cisplatin-resistant ovarian cancer cells [104,106]. Moreover, the small-molecule c-Myc inhibitor, 10058-F4, was reported to be effective in anti-cancer treatment, such as for leukemia [107], hepatocellular carcinoma [103], and prostate cancer [100]. These in vitro and in vivo observations suggest that c-Myc protects genomes of a tumor cell from therapeutic DNA-damaging drugs. In CD133-positive colon CSCs, knockdown of c-Myc expression upregulates the sensitivity to the anti-cancer drug treatment through the decrease in expression of ATP-binding cassette and multidrug resistance proteins, including ABCG2 and ABCB5 [108], implying that c-Myc might contribute to maintaining the chemoresistance of colon CSCs.

## 8. Roles of c-Myc in Colorectal Cancer Organoids

Recent biotechnology can transform human tissues into complex, valuable tissue products, such as organoids, which offer unique insights into the biological processes of the tissues. Three-dimensional (3D) organoid culture is derived from self-renewing stem cells, which can reproduce the original architecture in vivo as well as functions and genetic signatures. It also could be helpful for cancer research fields and personalized therapy [109].

The role of c-Myc in CRC organoids growth and proliferation are summarized below (Figure 5). In CRC, the ectopic expression of c-Myc by retrovirus rescues the reduction of organoid-forming efficiency and proliferation in Proliferating Cell Nuclear Antigen (PCNA)-associated factor (PAF) KO Lgr5^+^ organoids, suggesting that the PAF-transactivated c-Myc (PAF-Myc axis) is required for expansion of the intestinal stem and the progenitor cells during intestinal regeneration and tumorigenesis [110]. However, the ectopic expression of c-Myc could not recover the organoid development in PAF KO; Apc^Min/+^ state, suggesting that other additional pathways might be implicated in the PAF-influenced CRC stemness.

In the previous study, we established a novel 3D organoid culture model by an air-liquid interface (ALI) method, which contains many CSCs and exhibits resistance to anti-cancer drug treatment [111,112]. Hedgehog signals are implicated in the regulation of nuclear translocation of GLI-1 that triggers transcription of target genes, including c-Myc [113,114]. We demonstrated that combination therapy with agents that inhibit the Hedgehog signals, such as AY9944, GANT61, and 5-FU, irinotecan or oxaliplatin declined the cell viability of the CRC organoids when compared with the single treatment of each compound. Also, the expression of stem cell markers (c-Myc, CD44, and Nanog) was blocked by AY9944 or GANT61 treatment through reducing the expression of their transcription factor, GLI-1 [115,116]. These results suggest that Hedgehog signal mediates drug resistance of human colorectal tumor ALI organoids through regulation of the expression of stem cell markers, including c-Myc.

Moreover, Duong et al. [117] found that the cell migration-inducing and hyaluronan-binding protein (CEMIP) (endosomal protein)-dependent pathway maintains c-Myc protein levels through ERK1/2, which provides a metabolic advantage in selumetinib-resistant ex-vivo CRC organoids. Additionally, knockdown of the CEMIP gene circumvents resistance to MEK1 inhibition through a decrease in activation of ERK1/2 signaling and c-Myc [117].

In a recent study [118], a novel tumor suppressor gene, PR domain containing (PRDM)1, inhibited c-Myc-response genes and stem cell-related genes, which are necessary for the inhibition of proliferation and differentiation of primary CRC organoids [118]. Additionally, a recent study also showed that PRDM1 could inhibit SW620 colon cancer cell proliferation by inhibiting c-Myc [119].

Considering these reports, c-Myc might be essential for the generation and the proliferation of CRC organoids through the activation of various types of cancer stem cell-related signaling.

## 9. Therapeutic Opportunities Targeting c-Myc

A large number of proofs of principle experiments suggest that targeting c-Myc may have considerable therapeutic benefit in human tumors [120,121]. c-Myc binds to the MAX protein and forms dimers, which are necessary for the biological activation of c-Myc. Strategies have emerged to inhibit c-Myc expression, to interrupt c-Myc-Max dimerization, to inhibit c-Myc-Max DNA binding, to interfere with key c-Myc target genes, and to inhibit c-Myc in CSCs (Figure 6).

Recently, bromodomain and extra terminal domain (BET) family proteins have emerged as potent regulators of c-Myc expression in different tumors. Inhibition of BET with a drug-like molecule in multiple myeloma results in a remarkable decrease in c-Myc expression and cell death. It was also shown that several Burkitt lymphoma cell lines with c-Myc translocations are vulnerable to growth inhibition by BET inhibitors (JQ-1, OTX015, and TEN010). Inhibition of the BET/bromodomain-containing (BRD)4 protein is critical for transcription of the c-Myc promoter and was shown to be effective in an in vivo preclinical mode, implying that targeting c-Myc expression is possible in selected cancers [24,122].

The strategy focused on interrupting c-Myc-Max dimerization has been studied by several groups [123,124,125,126]. Although it has been documented that the inhibitors (10074-G5 and omomyc) are effective in vitro, evidence for in vivo effectiveness is still lacking. This means it can bring about good results when new chemistry is applied, such as click-chemistry, which links two moderate inhibitors against adjoining protein domains of the target to synthesize a more potent inhibitor.

Another strategy is to consider the transcriptional regulatory domain and the DNA binding domain as potential targets for small molecules (NY2279 and Mycrol1, 2, 3) that would nucleate polypeptide folding and close the domain in a non-functional way.

Other strategies have focused on targeting c-Myc target genes. For example, Kota et al. showed that administration of miR-26a suppresses tumorigenesis through inhibition of cell proliferation and induction of apoptosis in the c-Myc-induced liver cancer model mouse [127]. This strategy resulted in a remarkable response, suggesting that interfering with c-Myc regulated microRNAs could be therapeutically feasible [128,129]. c-Myc target genes such as lactate dehydrogenase A, ornithine decarboxylase, and glutaminase have also been challenged by small hairpin (sh)RNAs or drug-like small molecules in vivo [130,131,132]. Several ubiquitin ligase proteins such as F-box and WD repeat domain-containing (FBXW)7 target c-Myc for proteasomal degradation [133]. It was reported that FBXW7 is frequently mutated in CRC, which enhances the stability of c-Myc [134]. Ubiquitin specific peptidase (USP)28 antagonizes FBXW7-induced c-Myc ubiquitination. CRCs express high levels of USP28 that bind to FBXW7 and counteract its function. Deletion of USP28 diminishes c-Myc levels and prolongs life span in CRC models [135]. Enhancing c-Myc turnover might be a valid strategy to inhibit c-Myc function in colorectal cancer. These strategies mainly depend on the necessity of these target genes for the full transforming ability of c-Myc.

On the other hand, there are two major therapeutic strategies for targeting c-Myc in CSCs [136]. Firstly, c-Myc-dependent metabolic reprogramming is associated with CD44 variant-dependent redox stress regulation in CSCs. It was reported that c-Myc increases nicotinamide adenine dinucleotide phosphate hydrogenase (NADPH) production via the enhancement of glutaminolysis. Secondly, the FBW7-dependent c-Myc degradation pathway is also responsible for the chemotherapy resistance to the conventional treatments through inhibition of CSCs [136].

Collectively, new drugs targeting c-Myc and its target genes may be applied with molecular characterization of cancers for clinical classification and selection of combination therapies in the future. We should catch up with the current advances in the complex functions of c-Myc in cancer cells to realize precision medicine.

## 10. Conclusions

Myc family proteins are significant mediators of cell proliferation, apoptosis, and differentiation. Since deregulation of c-Myc genes contributes to the development of several cancers, c-Myc is an attractive target for cancer therapy. Identification of selective inhibitors of c-Myc is necessary for the establishment of more specific and less toxic therapeutic agents that could be used alone or in combination with conventional therapy. We discussed several promising strategies regarding c-Myc inhibition. Further investigations will provide new insights into the functions of c-Myc in tumorigenesis and may give the novel therapeutic agents for colorectal cancer.

## Figures and Tables

**Figure 1 ijms-20-02340-f001:**
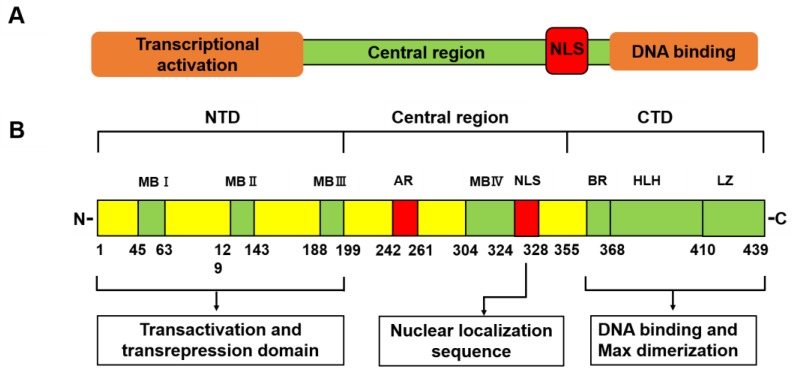
Structures of Myc family proteins (**A**). Mammalian Myc proteins consist of a transcriptional activation domain, a central region, a nuclear localization sequence (NLS), and an area involved in DNA binding. Structures of human c-Myc protein (**B**). The N-terminal transactivation domain (NTD) includes three conserved regions named as Myc box-I, II, and III (MBI, MBII, and MBIII) and the acidic region (AR). MBI and MBII are necessary for Myc transcriptional and cell-transforming activity as well as recruitment of Myc transactivation. MBIII regulates transcriptional suppression. In the central region, MBIV is involved in apoptosis, transformation and modulation of DNA-binding. The C-terminal domain (CTD) includes the basic, the helix-loop-helix, and the leucine zipper domains (b-HLH-LZ) and is necessary for dimerization with Myc associated protein X (MAX).

**Figure 2 ijms-20-02340-f002:**
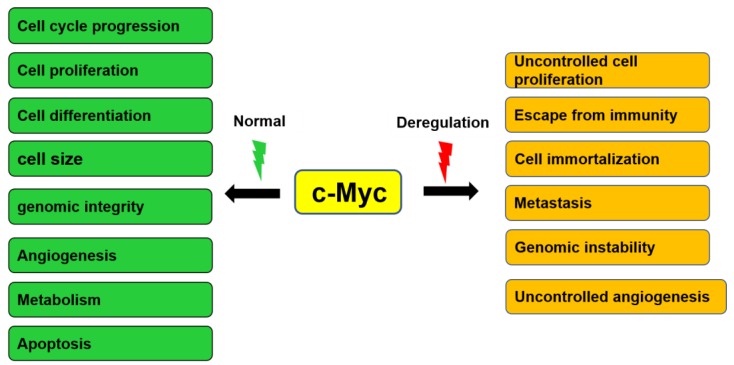
Cellular functions controlled by c-Myc during normal conditions and tumorigenesis. c-Myc mediates numerous biological effects, including cell cycle progression, cell proliferation, cell differentiation, cell size, genomic integrity, angiogenesis, cell metabolism, and apoptosis. Deregulation of c-Myc may result in uncontrolled cell proliferation, escape from immunity, cell immortalization, metastasis, genomic instability, and uncontrolled angiogenesis.

**Figure 3 ijms-20-02340-f003:**
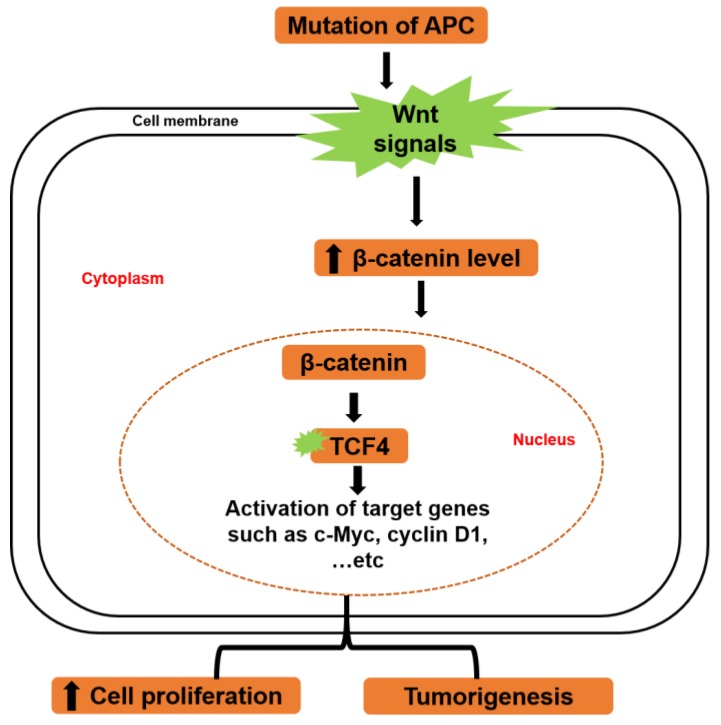
Regulation of Wnt/β-catenin/c-Myc signaling pathway. Mutation of adenomatous polyposis coli (APC) triggers Wnt signaling, which in turn increases the cytoplasmic level of free β-catenin migrating to the nucleus. In the nucleus, β-catenin activates transcriptional genes such as c-Myc and cyclin D1 through activation of transcription factor (TCF)4, which leads to the promotion of cell proliferation and tumorigenesis.

**Figure 4 ijms-20-02340-f004:**
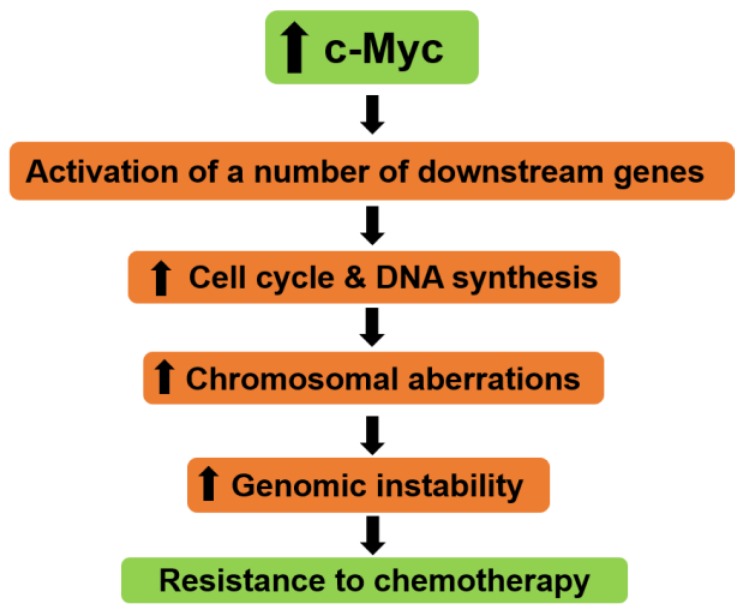
Roles of c-Myc in chemotherapy resistance. Overexpression of c-Myc may result in activation of several downstream genes, which leads to a promotion of cell cycle with DNA synthesis and an increase in chromosomal aberrations. These mechanisms ultimately initiate genomic instability and chemoresistance. The upper head arrows indicate increase while the lower ones indicate the consequences.

**Figure 5 ijms-20-02340-f005:**
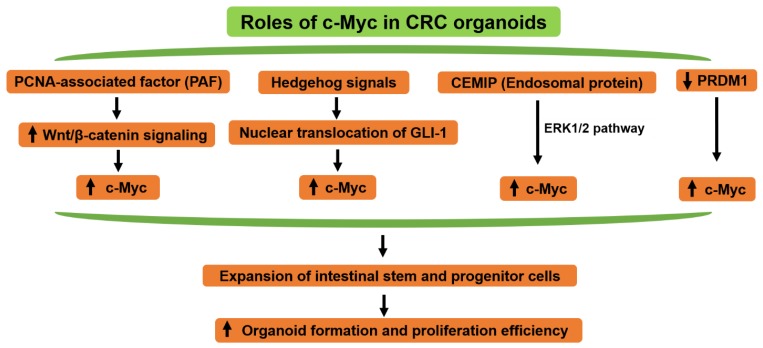
Roles of c-Myc in colorectal cancer (CRC) organoids. Overexpression of c-Myc results in expansion of the intestinal stem and the progenitor cells, which in turn increases organoid formation and proliferation. The mechanisms might be regulated by PCNA-associated factor **(**PAF)/Wnt/βcatenin signaling, Hedgehog signaling, cell migration-inducing and hyaluronan-binding protein (CEMIP)-dependent pathway through ERK1/2 or silencing the PR domain containing (PRDM)1 gene. The upper head arrows indicate increase while the lower ones indicate the consequences.

**Figure 6 ijms-20-02340-f006:**
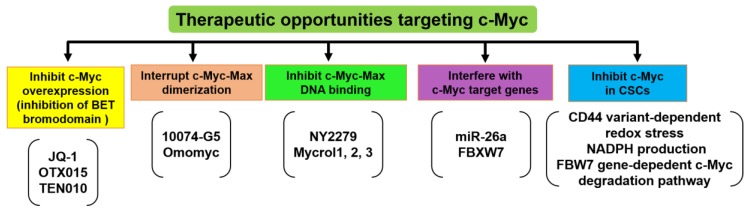
Strategies for targeting c-Myc. Several therapeutic agents were developed to inhibit c-Myc by different mechanisms. These include inhibitors that block Myc expression, including bromodomain and extra-terminal domain (BET) bromodomain inhibitors such as JQ-1, OTX015, and TEN010. Other inhibitors reduce c-Myc stability and functions by interfering with c-Myc-Max interaction (10074-G5 and omomyc) or inhibit its binding with DNA (NY2279 and Mycrol1, 2, 3). Some inhibitors interfere with c-Myc target genes such as miR-26a and F-box and WD repeat domain-containing (FBXW)7. In CSCs, inhibition of c-Myc-dependent metabolic reprogramming through CD44 variant-dependent redox stress, nicotinamide adenine dinucleotide phosphate hydrogenase (NADPH) production via decreasing glutaminolysis, or FBW7 gene-dependent c-Myc degradation pathway is regarded as a promising strategy.

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
