# Peer review of "Emerging Roles of C-Myc in Cancer Stem Cell-Related Signaling and Resistance to Cancer Chemotherapy: A Potential Therapeutic Target Against Colorectal Cancer"

_ijms, 2019, doi:10.3390/ijms20092340_

Round 1
Reviewer 1 Report
In this review, authors have discussed the role of c-MYC in colorectal CSCs, chemotherapy resistance and future therapeutic opportunities to target c-MYC. Authors have performed the study well. There are only few concerns this review as mentined below:
Authors can include therapeutic opportunities such as BET inhibitors to increase the abstract size.
Please mention any study which relate c-MYC overexpression with chemoresistance in CRC in the heading “Roles of c-Myc in resistance of chemotherapy”.
Please explain c-MYC-MAX in detail in heading 8 first para.
Author Response
Responses to Reviewer: 1
Q1. Authors can include therapeutic opportunities such as BET inhibitors to increase the abstract size.
A1. Thank you for giving us the valuable advice. Following the suggestion, we included the introduction of c-Myc inhibitors in the abstract (page 1, line 22).
Q2. Please mention any study which relate c-MYC overexpression with chemoresistance in CRC in the heading “Roles of c-Myc in resistance of chemotherapy”.
A2. Following the suggestion, we added the description about the relationship between c-Myc expression and chemoresistance in CRC (page 7, line 218-221). Thanks for the comment.
Q3. Please explain c-MYC-MAX in detail in heading 8 first para.
A3. Following the suggestions, we added the explanation on c-Myc-MAX in the first paragraph of chapter 8 (page 9, line 272-273). Thanks for the comment.
Reviewer 2 Report
The manuscript entitled “Emerging roles of c-Myc in cancer stem cell-related signaling and resistance to cancer chemotherapy: a potential therapeutic target against colorectal cancer” by Elbadawy et al. reviews some of the effects of c-Myc in colorectal cancer (CRC) and discusses the potential actionability of the gene from a therapeutic point of view. The review covers an interesting subject, but the authors are faced with a difficult question, because the subject has been often reviewed. After a brief Introduction, the authors review the structure of the proteins of MYC family, their general functions and the roles of c-Myc in cancer, with a brief note on CRC. Afterwards, the manuscript deals with the roles of c-Myc in the regulation of cancer stem cells, the resistance to chemotherapy and in CRC-derived organoids, to conclude with a section on the therapeutic possibilities involving targeting c-Myc.
The last four sections of the manuscript, especially those dealing with the CRC organoids and with the regulation of cancer stem cell-related signalling constitute the most original aspects of the review, because they have received less attention in previous similar reviews by other authors.
I think the interest of the manuscript should be greatly enhanced if the authors expand the most original sections mentioned above. Especially, the section dealing with CRC-derived organoids and with the conclusions on the function of c-Myc drawn from these studies might be enlarged, because the authors have made some personal contributions to this area. The proposed expansion may be done without enlarging the length of the manuscript, because some other sections, especially sections 1-3 may be abridged. This abridgment will agree with the title of the manuscript, which emphasizes the other issues. Also, some of the less informative figures (for instance, 2 and 4, whose content may be easily explained in the text) may be eliminated.
The text needs to be edited to improve the quality of English.
Finally, some other minor remarks:
1) There is a confusing statement on lines 44 ff. It is the gene and not the protein what is located on chromosome 8 and possesses introns and exons.
2) The paragraph starting on line 63 makes a doubtful interpretation of the results of Herbst et al. (ref. 32) especially in the commentary on proteolysis. Also, the expression “oncogenic transmutation” is not usual. It is better to say oncogenic transformation.
3) A reference for the experiments mentioned in lines 140-144 is needed.
4) Although it is not necessary, the manuscript will be improved by a mention to therapeutic approaches other than chemotherapy under section 6. I mean, for instance, the EGFR-directed drugs, such as cetuximab (see, for instance, the article by Yu et al. Oncotarget 2016 7(49):80888-80900).
5) The typo in the title of ref 33 when referring to G1/G2 must be corrected.
Author Response
Responses to Reviewer: 2
Q1. The last four sections of the manuscript, especially those dealing with the CRC organoids and with the regulation of cancer stem cell-related signaling constitute the most original aspects of the review, because they have received less attention in previous similar reviews by other authors. I think the interest of the manuscript should be greatly enhanced if the authors expand the most original sections mentioned above. Especially, the section dealing with CRC-derived organoids and with the conclusions on the function of c-Myc drawn from these studies might be enlarged, because the authors have made some personal contributions to this area. The proposed expansion may be done without enlarging the length of the manuscript, because some other sections, especially sections 1-3 may be abridged. This abridgment will agree with the title of the manuscript, which emphasizes the other issues. Also, some of the less informative figures (for instance, 2 and 4, whose content may be easily explained in the text) may be eliminated.
A1. Thank you for the comments indicating the novelty and originality of our manuscript and thanks again for giving us valuable advices. About enlarging the sections discussing the role of c-Myc in CRC organoids; unfortunately, the published data are few and we already referred to most of them in these sections. Also, we think that section 1-3 are necessary for readers to better understand the rest of the article and we already abridged them as much as possible. Figures 2 and 4 are summarizing in the general functions of c-Myc and its role in chemotherapy resistance, and we prefer to keep it in the article.
Q2. The text needs to be edited to improve the quality of English.
A2. Following the suggestion, we checked the grammar and spelling throughout the text again and carefully revised them.
Q3. There is a confusing statement on lines 44 ff. It is the gene and not the protein what is located on chromosome 8 and possesses introns and exons.
A3. Revised (page 2, line 46).
Q4. The paragraph starting on line 63 makes a doubtful interpretation of the results of Herbst et al. (ref. 32) especially in the commentary on proteolysis. Also, the expression “oncogenic transmutation” is not usual. It is better to say oncogenic transformation.
A4. Following the suggestion, we changed the sentence " c-Myc is assumed to be a critical part in the setting of gene expression, tuning c-Myc by proteolysis, and in oncogenic transmutation. " to " c-Myc is assumed to be a critical part in oncogenic transformation through the regulation of apoptotic responses [32] " (page 2, line 65-66).
Q5. A reference for the experiments mentioned in lines 140-144 is needed.
A5. Because we explained the role of Wnt/β-catenin, Hedgehog, and Notch pathways in CSCs in the following paragraph, we did not add a reference to this part. Thanks for the comment.
Q6. Although it is not necessary, the manuscript will be improved by a mention to therapeutic approaches other than chemotherapy under section 6. I mean, for instance, the EGFR-directed drugs, such as cetuximab (see, for instance, the article by Yu et al. Oncotarget 2016 7(49):80888-80900).
A6. Because we mainly focus on the c-Myc targeting drugs in the present review paper, we only selected these drugs and introduced in the section 6. Thanks for the comment.
Q7. The typo in the title of ref 33 when referring to G1/G2 must be corrected.
A7. Revised (page 12, line 420-421).
Reviewer 3 Report
Lines 295-297: It is not immediately apparent whether or not FBXW7 mutation and high levels of USP28 are related or separate events. Please make those statements more clear.
Fig6: Include the inhibitors available for the possible c-Myc targets in the figure.
Include information regarding the development phase of possible Myc pathway inhibitors-preclinical, Phase I, Phase II and their effectiveness at those stages.
Author Response
Responses to Reviewer: 3
Q1. Lines 295-297: It is not immediately apparent whether or not FBXW7 mutation and high levels of USP28 are related or separate events. Please make those statements more clear.
A1. We think FBXW7 mutation and high levels of USP28 are separate events. To avoid the confusion, we changed the sentence " Furthermore, CRC express high levels of ubiquitin specific peptidase (USP)28, a ubiquitin protease that binds to FBXW7 and counteracts its function; deletion of USP28 diminishes c-Myc levels and prolongs life span in CRC models [136]. " to " Ubiquitin specific peptidase (USP)28 antagonizes FBXW7-induced c-Myc ubiquitination. CRC express high levels of USP28 that binds to FBXW7 and counteracts its function. Deletion of USP28 diminishes c-Myc levels and prolongs life span in CRC models [136]" (page 10, line 301-303).
Q2. Fig6: Include the inhibitors available for the possible c-Myc targets in the figure.
A2. Revised (page 10, New Fig. 6).
Q3. Include information regarding the development phase of possible Myc pathway inhibitors-preclinical, Phase I, Phase II and their effectiveness at those stages.
A3. Thanks for the valuable advice. We surely agree that the information on the preclinical phase and effectiveness at each stage is essential for considering the therapeutic strategies. However, to introduce the basic strategies for inhibiting c-Myc pathway and also to avoid the confusion of readers, we did not describe the detailed preclinical information on each inhibitor in this review article.
Round 2
Reviewer 2 Report
After the fast revision made by the authors, the manuscript entitled “Emerging roles of c-Myc in cancer stem cell-related signaling and resistance to cancer chemotherapy: a potential therapeutic target against colorectal cancer” by Elbadawy et al. seems improved. The authors have dealt with the questions posed by the three reviewers and they have introduced most of the changes suggested by us. It is a pity they are not able to enlarge the section dealing with CRC-derived organoids because, as I mentioned in my first review, it is the most original part of the manuscript.
My former comments on shortening some sections and on the possibility of eliminating two figures were suggested to be done in the event that the authors would enlarge other sections. In view of the difficulty to do this, as the authors mention, their rebuttal to address those comments is reasonable.
This manuscript is a resubmission of an earlier submission. The following is a list of the peer review reports and author responses from that submission.